# *Ficus carica* Fruits, By-Products and Based Products as Potential Sources of Bioactive Compounds: A Review

Candela Teruel-Andreu, Lucía Andreu-Coll, David López-Lluch [ID], Esther Sendra [ID], Francisca Hernández [ID] and Marina Cano-Lamadrid *[ID]

Centro de Investigación e Innovación Agroalimentaria y Agroambiental (CIAGRO-UMH), Miguel Hernández University, Ctra. Beniel, Km 3.2, 03312 Orihuela, Spain; candela.teruel@goumh.umh.es (C.T.-A.); lucia.andreu1@gmail.com (L.A.-C.); david.lopez@umh.es (D.L.-L.); esther.sendra@umh.es (E.S.); Francisca.hernandez@umh.es (F.H.)
* Correspondence: marina.canol@umh.es

**Abstract:** In this review, studies (*n* = 41) were searched in which the compounds and contents were determined for whole fig fruit, peel, leaves and pulp, the types of fig-based products were identified and their total phenols and antioxidant capacity as well as the potential uses of different extracts of fig parts were analyzed. There is a need to reduce the fruit's environmental impacts (zero waste), and bioactive compounds from fig fruits present a high added value as functional ingredients. Focusing on fig by-products (peel, seeds, no-optimal fruits and leaves), individual compounds and/or extracts can increase the functional, nutritional and techno-functional properties of food products such as additives. A high number of phenolic compounds was found in whole fruit (*n* = 19), peel (*n* = 26), pulp (*n* = 24) and leaves (*n* = 42). Quercetin-3-O-rutioside was reported as the major individual phenolic compound in whole figs, while cyanidin-3-rutinoside, epicatechin and caftaric acid were the highest compounds in peel, pulp and leaves, respectively. A potential strategy could be the development of novel additives and/or ingredients for food industry from fig by-products. Therefore, the use and valorization of the waste material produced during fig processing should be further investigated.

**Keywords:** figs; leaf; revalorization; pulp; added-value

## 1. Introduction

*Ficus carica* L. (fig) is a species of the very large number of the genus *Ficus* belonging to the Moraceae famil, characterized by milky latex in all parenchymatous tissue, unisexual flowers, anatropous ovules and aggregated drupes or achenes [1]. Figs are infruitescences—the true fruits of the fig are located inside the fig or siconio, which are called achenes [2]. The fleshy and sweet part of the fig corresponds to the flower receptacles that, after fertilization, become swollen and fleshy [3]. It is worth mentioning that it is one of the oldest species domesticated by humanity [4]. In the Middle East and the Mediterranean region, the fig has been included in the diet since ancient times and is considered a symbol of health [5]. It has been suggested that the cultivation of the fig originated in the East Mediterranean region, which was later expanded into the West Mediterranean area [6].

According to the Food and Agriculture Organization (FAO) of the United Nations, the world production of fig fruit is stable. Worldwide, the area under cultivation of fig trees exceeds 289,818 ha, with an estimated production of 1,315,588 t [7]. Turkey is the biggest world producer, with 310,000 t in 2019, followed by Egypt, Morocco, Iran, Algeria and Spain. Therefore, of utmost importance for fig production is still the Mediterranean basin and the Near East [7]. Spain is the main source of figs in Europe (51,600 t), followed by Greece (19,730 t) and Italy (11,830 t) [7]. In Spain, the main producer is Extremadura (37,382 t), followed by Cataluña (5834 t) and the Comunidad Valenciana (2932 t) [8]. Because the fig tree is highly resistant to salinity and active calcium, it is quite suitable for marginal areas, such as southeastern Spain [9]. Taking production, yield and size of the cultivars cultivated

in Spain into account, "Banane" and "Brown Turkey" are the main cultivars [10]. On the other hand, other authors indicated that the most important cultivars are "Cuello Dama Blanco" and "Colar de Elche", because they exhibit the best organoleptic punctuation due to their higher content of sugars [11].

A high number of bioactive compounds have been found in the peel, flesh, leaves and whole fruits of figs, such as cyanidin, chlorogenic acid, rutin, luteolin and (+)-Catechin, among others [2,12,13]. Several authors have indicated that these compounds present potential health properties, such as antibacterial, hepatoprotective, antidiabetic, anti-inflammatory, antioxidant and anticancer activity [14–16]. Therefore, consumer demand for fig fruit and fig-based products has increased in the past decades [17]. It is essential to highlight that no health claim is yet authorized for "antioxidants", "anthocyanins" or "fig" by the European Food Safety Agency (EFSA). There is just one authorized claim (for polyphenols): hydroxytirosol and derivatives in olive oil. On the other hand, the high concentration of calcium (133 mg/100 g) [1] in fig fruits allows to mention the nutritional claim "rich in calcium", because its content is higher than 10% of the RDI (recommended dietary intake).

Fig products have been used in traditional medicine to treat many diseases, mainly in the dermatological field [18]. Abbasi et al. [19] studied the application of fig plant extracts and showed their effectiveness in relieving symptoms of atopic dermatitis, and can, hence, be used instead of cortisones. Moreover, another study reported the potential of fig plant extract to be used as a treatment of and prevention for skin warts and cervical cancer [20]. Ongoing research suggests anticancer effects of two components of fig leaf extract, bergapten and psoralen, which could be a good source for developing drugs to suppress the growth of cancer cells [18]. Additionally, other studies concluded that figs are a concentrated source of benzaldehyde [21]. There are studies that also show the potential of fig extracts to produce medicines for cardiovascular diseases, by the content in components such as flavone, rutine and quercetin [22]. Additionally, fig fruits as well as leaves have a high nutritional value and their high content of dietary fiber is widely known [22]. Constipation is a very common health problem, and laxative foods such as figs and their derivatives could be considered effective for this problem [23]. Similar results were also reported in a previous study, in which fig leaf extracts were used to help combat eating and lifestyle disorders [24]. Additionally, Ajmal et al. [25] recognized the efficacy of fig leaf extracts for reducing blood glucose levels.

On the other hand, there is a need to reduce the fig's environmental impacts (zero waste) and fruit-based products present a high added value as functional ingredients. Among fruit-based products, peels, seeds, no-optimal fruits and leaves, among others, can be found. Focusing on fig by-products, the peel and leaf extracts could increase the nutritional and pharmaceutical properties of food products such as additives [4]. Therefore, there is a great need to generate comprehensive information about the bioactive compounds of fig fruits, their derivatives/by-products (peel, leaves and oil) and fig-based products. How the processing (drying and preparation of jams) and storage have affected the phenolic composition of fig products will also be an objective of this review.

## 2. Scientific Literature Review

This review is organized as a research paper. A scoping review was used to synthesize the evidence and assess the scope of the 41 studies on the topic. This review was based on the PRISMA Extension (PRISMA-ScR) approach [26] for Scoping Reviews. A comprehensive literature search—Scopus and ScienceDirect—was performed in August 2021 and was limited to articles published in English since 1990 (Figure 1). Text words and controlled vocabulary for several concepts (*Ficus carica,* by-products, bioactive compounds, fig, peel, leaves and revalorization) within the titles, abstracts and keywords were used. The main focus has been given to studies published in journals included in the Journal Citation Reports. Only research papers that included the experimental design and data treatment were selected. The structure of the review allows a dissection of (i) which compounds

and their content in whole fig fruit, peel, leaves and pulp (*n* = 12); (ii) types of fig-based products and their total phenols and antioxidant capacity (*n* = 16); and (iii) uses of different extracts of fig parts (*n* = 13).

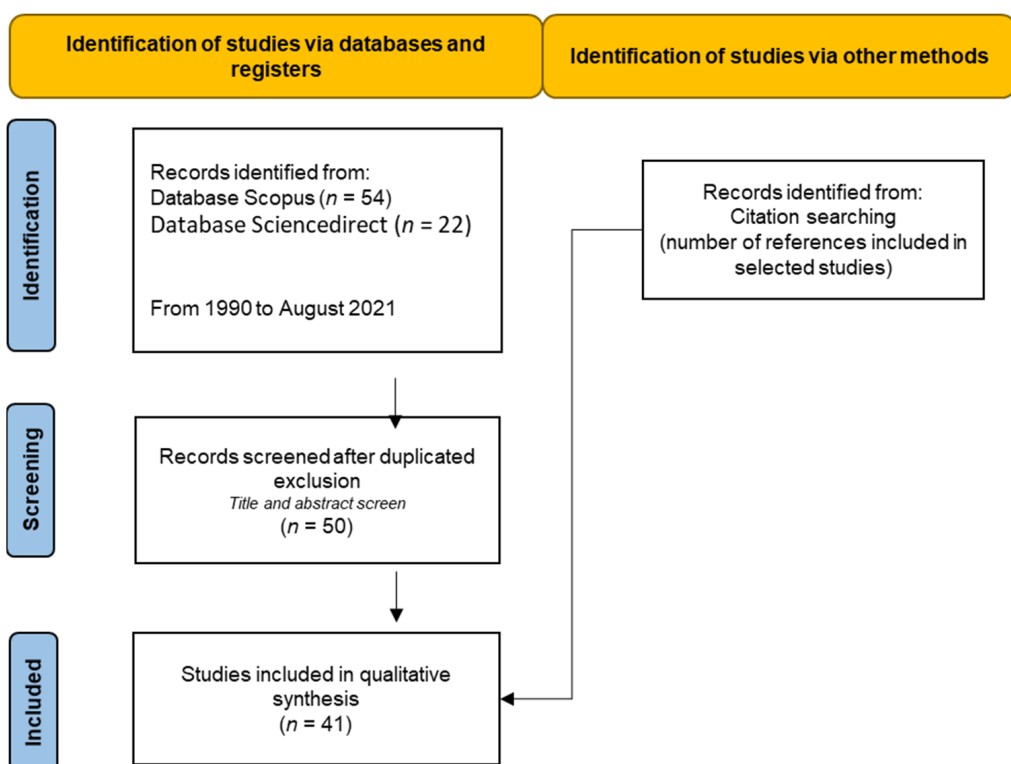

**Figure 1.** Flow diagram describing the study selection process of the scientific literature.

## 3. Bioactive Compounds in Different Fig Parts

Table 1 shows the individual phenolic compounds found in different part of fig parts: whole fruit (*n* = 19), peel (*n* = 21), pulp (*n* = 22) and leaves (*n* = 40). Identified compounds belong to different chemical families, such as phenolic acids (caffeic acid, chlorogenic acid, ferulic acid, coumaric acid, syringic acid, quinol and gallic acid) and flavonoids (catechin, kaempferol, quercitin and myricetin). The chemical structure of the main compounds found in fig fruits and derivatives are shown in recent studies related to chemical composition [14,16]. It is important to highlight that the results in the literature are not always reported in a unified manner, which makes it difficult to compare research findings. Data are expressed as dried matter/weight (dw) and fresh matter/weight (fw). Depending on the part of the fruit, maturity index, variety and region, the type of compounds varies. Quercetin-3-O-rutioside was reported as the major individual phenolic compound in whole figs (Table 1), followed by polymeric procyanidins, quercentin-3-glucoside, chlorogenic acid and cyanidin-3-0-rutinoside. As for the peel's bioactive compounds, cyanidin-3-rutinoside was the most abundant, followed by cyanidin-3,5-diglucoside, cyanidin-3-O-diglucoside, epitecatechin, catechin and quercetin-rutinoside. Epicatechin and cyanidin-3-rutinoside were the main compounds found in fig pulp, while caftaric acid, in the form of kaempferol 3-O-glucoside, was the main compound reported in fig leaves. On the other hand, Badgujar et al. [15] and Li et al. [14] reviewed the phytochemical composition of *Ficus carica* fruits and their derivatives. This study only indicated the profile of the bioactive compounds (isolation of phytosterols, anthocyanins, phenolic components and a few other classes of secondary metabolites), not the quantification. Therefore, these manuscripts were not added to Table 1. Most of these phytochemicals were found in latex, followed by leaves, fruit and root. Additionally, Li et al. [14] collected data of the phytochemical composition related to health properties, indicating that conventional and modern isolation and characterization techniques were used for the identification of about 126 chemical constituents, which were

divided into eight categories: hydroxybenzoic acids, hydroxycinnamic acids, flavonoids, coumarins, furanocoumarins, volatile constituents, triterpenoids and miscellaneous [14].

**Table 1.** Bioactive compounds and their content (minimum and maximum) in fig whole fruits, peel, pulp and leaves.

| Plant Part | Compound | Minimun Value | Maximum Value | Unit | Reference |
|---|---|---|---|---|---|
| **Whole figs** | | | | | |
| | Catechin | 0.2060 | 0.9570 | (mg g$^{-1}$ dw) | [12] |
| | | 0.0127 | 0.1670 | (mg g$^{-1}$ fw) | [6] |
| | Epicatechin | 0.0900 | 0.4310 | (mg g$^{-1}$ dw) | [12] |
| | | 0.0058 | 0.3210 | (mg g$^{-1}$ fw) | [6] |
| | Polymeric procyanidins | 0.5560 | 2.6800 | (mg g$^{-1}$ dw) | [12] |
| | Rutin | 0.0489 | 0.2870 | (mg g$^{-1}$ fw) | [27] |
| | | 0.0089 | 0.2870 | (mg g$^{-1}$ fw) | [6] |
| | Cyanidin-3,5-*O*-diglucoside | 0.0000 | 0.0190 | (mg g$^{-1}$ dw) | [12] |
| | Cyanidin-3-*O*-rutinoside | 0.0040 | 1.1620 | (mg g$^{-1}$ dw) | [12] |
| | Pelargonidin-3-*O*-rutinoside | 0.0000 | 0.0380 | (mg g$^{-1}$ dw) | [12] |
| | Chlorogenic acid | 0.0880 | 1.2450 | (mg g$^{-1}$ dw) | [12] |
| | | 0.0105 | 0.0157 | (mg g$^{-1}$ fw) | [6] |
| | keampferol-3-glucoside | | 0.0013 | (mg g$^{-1}$ fw) | [6] |
| | keampferol-3-*O*-rutinoside | 0.0060 | 0.3050 | (mg g$^{-1}$ dw) | [12] |
| | Quercetin 3-glucoside | 0.0041 | 1.4020 | (mg g$^{-1}$ fw) | [6] |
| | Quercetin-3-*O*-rutioside | 0.3990 | 3.2830 | (mg g$^{-1}$ dw) | [12] |
| | Quercetin-3-galactoside | 0.0460 | 0.1420 | (mg g$^{-1}$ dw) | [12] |
| | Quercetin-3-*O*-malonyl-galactoside | 0.0530 | 0.5800 | (mg g$^{-1}$ dw) | [12] |
| | Apigenin-*C*-hexoside-pentoside | 0.0050 | 0.2430 | (mg g$^{-1}$ dw) | [12] |
| | Gallic acid | 0.0010 | 0.0038 | (mg g$^{-1}$ fw) | [27] |
| | | 0.0030 | 0.0280 | (mg g$^{-1}$ fw) | [6] |
| | Syringic acid | 0.0002 | 0.0010 | (mg g$^{-1}$ fw) | [27] |
| | Ellagic acid | | 0.0020 | (mg g$^{-1}$ fw) | [6] |
| | Syringic acid | 0.0003 | 0.0008 | (mg g$^{-1}$ fw) | [6] |
| **Peel** | | | | | |
| | Catechin | 0.0220 | 0.2060 | (mg g$^{-1}$) | [10] |
| | | 0.0009 | 0.0239 | (mg g$^{-1}$ dw) | [28] |
| | Epicatechin | 0.0350 | 0.2570 | (mg g$^{-1}$) | [10] |
| | | 0.0027 | 0.0547 | (mg g$^{-1}$ dw) | [28] |
| | (epi)catechin-(4-8)-Cy 3-rutinoside | 0.0004 | 0.0009 | (mg g$^{-1}$ fw) | [29] |
| | Carboxypyrano-Cy 3-rutinoside | 0.0005 | 0.0013 | (mg g$^{-1}$ fw) | [29] |
| | Cyanidin-3,5-diglucoside | 0.0020 | 0.0052 | (mg g$^{-1}$ fw) | [29] |
| | | 0.0008 | 0.4941 | (mg g$^{-1}$ dw) | [28] |
| | Cyanidin-3-malonylglycosyl-5-glucoside | 0.0005 | 0.0014 | (mg g$^{-1}$ fw) | [29] |
| | Cyanidin-3-glucoside | 0.0015 | 0.0154 | (mg g$^{-1}$ fw) | [29] |
| | | 0.1100 | 0.0060 | (mg g$^{-1}$ fw) | [2] |
| | | 0.0001 | 0.0083 | (mg g$^{-1}$) | [10] |
| | Cyanidin-3-rutinoside dimer | 0.0004 | 0.0009 | (mg g$^{-1}$ fw) | [29] |
| | Cyanidin-3-malonylglucoside | 0.0006 | 0.0035 | (mg g$^{-1}$ fw) | [29] |
| | Pg 3-rutinoside | 0.0005 | 0.0035 | (mg g$^{-1}$ fw) | [29] |
| | Cyanidin-3-*O*-rutinoside | 0.0079 | 0.1050 | (mg g$^{-1}$) | [10] |
| | | 0.0154 | 0.0783 | (mg g$^{-1}$ fw) | [29] |
| | | 0.2410 | 1.0890 | (mg g$^{-1}$ fw) | [2] |
| | | 0.0008 | 0.4787 | (mg g$^{-1}$ dw) | [28] |
| | Pelargonidin-3-*O*-rutinoside | 0.0000 | 0.0107 | (mg g$^{-1}$) | [10] |
| | | 0.0042 | 0.0126 | (mg g$^{-1}$ dw) | [28] |
| | Chlorogenic acid | 0.0005 | 0.0088 | (mg g$^{-1}$ dw) | [28] |
| | | 0.0200 | 0.0580 | (mg g$^{-1}$ fw) | [2] |
| | | 0.0020 | 0.0260 | (mg g$^{-1}$) | [10] |
| | Luteolin-7- OGlucoside | 0.0019 | 0.0179 | (mg g$^{-1}$ dw) | [28] |
| | Luteolin 6C-hexose-8C-pentose | 0.0010 | 0.0190 | (mg g$^{-1}$ fw) | [2] |
| | Kaempferol-rutinoside | 0.0020 | 0.0070 | (mg g$^{-1}$ fw) | [2] |

**Table 1.** *Cont.*

| Plant Part | Compound | Minimun Value | Maximum Value | Unit | Reference |
|---|---|---|---|---|---|
| | Quercetin | 0.0009 | 0.0595 | (mg g$^{-1}$ dw) | [28] |
| | Quercetine-acetilglucoside | 0.0020 | 0.0170 | (mg g$^{-1}$ fw) | [2] |
| | Quercetin-rutinoside | 0.0290 | 0.1580 | (mg g$^{-1}$ fw) | [2] |
| | Quercetine-glucoside | 0.0020 | 0.0320 | (mg g$^{-1}$ fw) | [2] |
| | Ellagic acid | 0.0150 | 0.0330 | (mg g$^{-1}$) | [10] |
| **Pulp** | | | | | |
| | Catechin | 0.0140 | 0.0670 | (mg g$^{-1}$) | [10] |
| | Epicatechin | 0.0140 | 0.1330 | (mg g$^{-1}$) | [10] |
| | (epi)catechin-(4-8)-Cy 3-glucoside | 0.0000 | 0.0001 | (mg g$^{-1}$ fw) | [29] |
| | (epi)catechin-(4-8)-Cy 3-rutinoside | 0.0000 | 0.0006 | (mg g$^{-1}$ fw) | [29] |
| | Cyanidin-3,5-diglucoside | 0.0001 | 0.0006 | (mg g$^{-1}$ fw) | [29] |
| | Cyanidin-33-malonylglycosyl-5-glucoside | 0.0000 | 0.0001 | (mg g$^{-1}$ fw) | [29] |
| | Cyanidin-33-glucoside | 0.0005 | 0.0022 | (mg g$^{-1}$ fw) | [29] |
| | Cyanidin-33-rutinoside | 0.0045 | 0.0102 | (mg g$^{-1}$ fw) | [29] |
| | Cyanidine-3-*O*Rutinoside | 0.0008 | 0.0105 | (mg g$^{-1}$ dw) | [28] |
| | Cyanidin-33-malonylglucoside | 0.0000 | 0.0001 | (mg g$^{-1}$ fw) | [29] |
| | Carboxypyrano-Cy 3-rutinoside | 0.0000 | 0.0009 | (mg g$^{-1}$ fw) | [29] |
| | Pelargonidin-3-rutinoside | 0.0000 | 0.0001 | (mg g$^{-1}$ fw) | [29] |
| | Pn 3-rutinoside | 0.0000 | 0.0001 | (mg g$^{-1}$ fw) | [29] |
| | Chlorogenic acid | 0.0010 | 0.0130 | (mg g$^{-1}$ fw) | [2] |
| | | 0.0010 | 0.0140 | (mg g$^{-1}$) | [10] |
| | Quercetinrutinoside | 0.0040 | 0.0170 | (mg g$^{-1}$ fw) | [2] |
| | Cyanidin-3-rutinoside | 0.0100 | 0.0950 | (mg g$^{-1}$ fw) | [2] |
| | Cyanidin-3-*O*-glucoside | 0.0001 | 0.0083 | (mg g$^{-1}$) | [10] |
| | Cyanidin-3-*O*-rutinoside | 0.0079 | 0.1050 | (mg g$^{-1}$) | [10] |
| | Pelargonidin-3-*O*-rutinoside | 0.0001 | 0.0107 | (mg g$^{-1}$) | [10] |
| | Quercitin-3-*O*-rutinoside | 0.0010 | 0.0190 | (mg g$^{-1}$) | [10] |
| | Quercitin-3-acetylglucoside | 0.0010 | 0.0140 | (mg g$^{-1}$) | [10] |
| | Ellagic acid | 0.0070 | 0.0140 | (mg g$^{-1}$) | [10] |
| **Leaf** | | | | | |
| | (+)-catechin | 0.5200 | 0.7400 | (mg g$^{-1}$ dw) | [30] |
| | Caffeoylmalic acid | 0.7900 | 5.9700 | (mg g$^{-1}$ dw) | [30] |
| | | 1.3860 | 7.4650 | (mg g$^{-1}$ dw) | [13] |
| | *p*-Coumaroyl derivative | 0.3920 | 0.7130 | (mg g$^{-1}$ dw) | [13] |
| | *p*-Coumaroylquinic acid | 0.3500 | 1.3710 | (mg g$^{-1}$ dw) | [13] |
| | *p*-Coumaroyl malic acid | 0.3380 | 0.7740 | (mg g$^{-1}$ dw) | [13] |
| | Caffeic acid derivates | 0.4240 | 0.5920 | (mg g$^{-1}$ dw) | [13] |
| | Caffeic acid | | 2.4800 | (mg g$^{-1}$ dw) | [31] |
| | Isoschaftoside | 0.1420 | 0.9910 | (mg g$^{-1}$ dw) | [13] |
| | Schaftoside | 0.0940 | 0.5180 | (mg g$^{-1}$ dw) | [13] |
| | Kampherol | | 0.8800 | (mg g$^{-1}$ dw) | [31] |
| | kaempferol 3-*O*-glucoside (astragalin) | 12.4300 | 22.7000 | (mg g$^{-1}$ dw) | [30] |
| | | 0.0400 | 0.3890 | (mg g$^{-1}$ dw) | [13] |
| | Kaempferol derivative | 0.0190 | 0.0600 | (mg g$^{-1}$ dw) | [13] |
| | Quercitin | | 13.4000 | (mg g$^{-1}$ dw) | [31] |
| | Quercetin derivative | 0.0480 | 0.2110 | (mg g$^{-1}$ dw) | [13] |
| | | 3.7200 | 7.5100 | (mg g$^{-1}$ dw) | [30] |
| | Rutin (quercetin-3-*O*-rutinoside) | 0.0097 | 0.6874 | (mg g$^{-1}$ fw) | [27] |
| | | 1.6480 | 8.2180 | (mg g$^{-1}$ dw) | [13] |
| | Quercetin 3-*O*-glucoside (isoquercetin) | 5.3600 | 12.4500 | (mg g$^{-1}$ dw) | [30] |
| | Quercetin 3-*O*-malonyl-glucoside | 0.1640 | 2.6210 | (mg g$^{-1}$ dw) | [13] |
| | Isoquercetin | 0.0760 | 1.5460 | (mg g$^{-1}$ dw) | [13] |
| | Gallic acid | | 1.5000 | (mg g$^{-1}$ dw) | [31] |
| | Psolaren | 0.3620 | 1.4920 | (mg g$^{-1}$ dw) | [13] |
| | Bergapten (5 methoxypsolaren) | 0.4450 | 0.6270 | (mg g$^{-1}$ dw) | [13] |
| | Psolaric acid isobar | 0.3440 | 0.4710 | (mg g$^{-1}$ dw) | [13] |

**Table 1.** *Cont.*

| Plant Part | Compound | Minimun Value | Maximum Value | Unit | Reference |
|---|---|---|---|---|---|
| | 3-*O*-caffeoylquinic | | 0.0020 | (mg g$^{-1}$ dw) | [31] |
| | acid (chlorogenic acid) | 1.3100 | 3.5400 | (mg g$^{-1}$ dw) | [30] |
| | | 0.3400 | 0.5900 | (mg g$^{-1}$ dw) | [30] |
| | 5-*O*-caffeoylquinic acid | 0.4050 | 2.0610 | (mg g$^{-1}$ dw) | [13] |
| | | 0.4736 | 1.158.8 | (mg g$^{-1}$ dw) | [32] |
| | Ferulic acid | | 0.0320 | (mg g$^{-1}$ dw) | [31] |
| | | | 11.9838 | (mg g$^{-1}$ dw) | [32] |
| | Pyrogallol | | 0.0060 | (mg g$^{-1}$ dw) | [31] |
| | Quinol | | 0.0110 | (mg g$^{-1}$ dw) | [31] |
| | *p*-Hydroxy benzoic acid | | 3.5000 | (mg g$^{-1}$ dw) | [31] |
| | Dihydroxybenzoic acid | 1.1500 | 2.1500 | (mg g$^{-1}$ dw) | [30] |
| | Vanillic acid | | 0.0790 | (mg g$^{-1}$ dw) | [31] |
| | Syringic acid | | 0.0970 | (mg g$^{-1}$ dw) | [31] |
| | *o*-Coumaric acid | | 0.0110 | (mg g$^{-1}$ dw) | [31] |
| | *p*-Coumaric acid | | 0.0130 | (mg g$^{-1}$ dw) | [31] |
| | Benzoic acid | | 0.3200 | (mg g$^{-1}$ dw) | [31] |
| | Caftaric acid | | 40.2000 | (mg g$^{-1}$ dw) | [31] |
| | Ellagic acid | | 0.5240 | (mg g$^{-1}$ dw) | [31] |
| | Salicylic acid | | 0.0450 | (mg g$^{-1}$ dw) | [31] |
| | Myricetin | | 0.4140 | (mg g$^{-1}$ dw) | [31] |
| | Rosmarinic acid | | 0.2700 | (mg g$^{-1}$ dw) | [31] |
| | Ligstroside | | 0.1880 | (mg g$^{-1}$ dw) | [31] |

## 4. Bioactive Content of Fig-Based Products and Their Antioxidant Activity

In general, fig fruits have mainly been consumed fresh and dried, but they have also traditionally been preserved and processed into jams [21]. Nowadays, consumer trends have changed and there is an increase in the range of other products based on figs [4]. Table 2 shows important information (type, cultivar and treatment) and bioactive compounds (total phenols, total flavonoids and total anthocyanins) as well as the antioxidant capacity of products based on fig fruits and their by-products. Additionally, the following lines include more information on the main reported fig-based products.

**Table 2.** Fig cultivar, total phenols, total flavonols and total anthocyanins, and the antioxidant capacity of the reported fig-based products.

| Sample | Variety/Origin | Treatment | Total Phenols | Total Flavonoids | Total Anthocyanins | Antioxidative Capacity | | References |
|---|---|---|---|---|---|---|---|---|
| Fig jam | Khudeiri | 0 [a] | 291.42 ± 44.9 (mg GAE kg$^{-1}$) | nd | 16.45 ± 1.2 (mg cya-3-glu kg$^{-1}$) | DPPH | 15.52 ± 0.5 (%) | [33] |
| Fig jam | Khudeiri | 1 [a] | 235.45 ± 2.6 (mg GAE kg$^{-1}$) | nd | 13.70 ± 1.0 (mg cya-3-glu kg$^{-1}$) | DPPH | 13.71 ± 0.3 (%) | [33] |
| Fig jam | Khudeiri | 2 [a] | 233.57 ± 0.5 (mg GAE kg$^{-1}$) | nd | 13.80 ± 1.1 (mg cya-3-glu kg$^{-1}$) | DPPH | 15.11 ± 2.0 (%) | [33] |
| Fig jam | Khudeiri | 3 [a] | 140.30 ± 5.3 (mg GAE kg$^{-1}$) | nd | 11.95 ± 1.7 (mg cya-3-glu kg$^{-1}$) | DPPH | 13.52 ± 0.4 (%) | [33] |
| Fig jam | Khudeiri | 4 [a] | 145.90 ± 13.2 (mg GAE kg$^{-1}$) | nd | 13.45 ± 0.1 (mg cya-3-glu kg$^{-1}$) | DPPH | 12.35 ± 0.5 (%) | [33] |
| Fig jam | Khudeiri | 5 [a] | 130.97 ± 2.6 (mg GAE kg$^{-1}$) | nd | 11.20 ± 0.6 (mg cya-3-glu kg$^{-1}$) | DPPH | 8.96 ± 2.1 (%) | [33] |
| Dry figs | "Bela petrovka" | Drying oven | 530.2 mg (GAE kg$^{-1}$ dw) | nd | nd | | nd | [34] |
| Dry figs | Serbia | | 195.33 ± 1.07 (mg/100 g dm) | nd | nd | ABTS | 0.388 ± 0.042 (mmol/100 g DM) | [35] |
| Dry figs | Turkey | | 19.2 (mg/100 g fw) | nd | nd | | nd | [2] |
| Dry figs | Spain | | 19.1 (mg/100 g fw) | nd | nd | | nd | [2] |
| Dry figs | Cuello dama | | 17.8 (mg/100 g fw) | nd | nd | | nd | [2] |
| Dry figs | Saoudi douiret | Direct solar dryer | 201.76 mg (GAE/100 g DM) | 112.28 mg (QE/100g DM) | nd | DPPH | 418.51 mg (TEAC/100 g DM) | [36] |
| Dry figs | Bayoudhi douiret | Direct solar dryer | 73.74 mg (GAE/100 g DM) | 57.96 mg (QE/100g DM) | nd | DPPH | 131.55 mg (TEAC/100 g DM) | [36] |
| Dry figs | Mission | Freeze drying [b] | 3.08 ± 0.4 (mg CEg$^{-1}$) | nd | nd | DPPH | 2.0 ± 0.3 (μM eq trolox g$^{-1}$) | [37] |
| Dry figs | Mission | Drying 45 °C [b] | 3.35 ± 0.2 (mg CEg$^{-1}$) | nd | nd | DPPH | 3.4 ± 0.3 (μM eq trolox g$^{-1}$) | [37] |
| Dry figs | Mission | Drying 55 °C [b] | 3.23 ± 0.3 (mg CEg$^{-1}$) | nd | nd | DPPH | 3.7 ± 0.2 (μM eq trolox g$^{-1}$) | [37] |
| Dry figs | Mission | Drying 65 °C [b] | 3.72 ± 0.2 (mg CEg$^{-1}$) | nd | nd | DPPH | 3.8 ± 0.3 (μM eq trolox g$^{-1}$) | [37] |
| Dry figs | Mission | Freeze drying [c] | 3.08 ± 0.4 (mg CEg$^{-1}$) | nd | nd | DPPH | 2.0 ± 0.3 (μM eq trolox g$^{-1}$) | [37] |
| Dry figs | Mission | Drying 45 °C [c] | 2.62 ± 0.2 (mg CEg$^{-1}$) | nd | nd | DPPH | 3.5 ± 0.3 (μM eq trolox g$^{-1}$) | [37] |
| Dry figs | Mission | Drying 55 °C [c] | 3.13 ± 0.3 (mg CEg$^{-1}$) | nd | nd | DPPH | 3.4 ± 0.3 (μM eq trolox g$^{-1}$) | [37] |
| Dry figs | Mission | Drying 65 °C [c] | 4.73 ± 0.7 (mg CEg$^{-1}$) | nd | nd | DPPH | 3.4 ± 0.7 (μM eq trolox g$^{-1}$) | [37] |
| Fermented figs | Mission | | 4.77 (mg GAE/g of dm) | nd | nd | DPPH | 0.53 (mg of GAE/g of dm) | [38] |
| Biscuit | Turkey | 5% Fig seed [d] | 145.28 ± 0.34 (mg GAE/100 g) | nd | nd | DPPH | 10.36 ± 0.04 (%) | [39] |
| Biscuit | Turkey | 10% Fig seed [d] | 163.21 ± 0.16 (mg GAE/100 g) | nd | nd | DPPH | 17.48 ± 0.09 (%) | [39] |
| Biscuit | Turkey | 15% Fig seed [d] | 76.84 ± 0.44 (mg GAE/100 g) | nd | nd | DPPH | 25.36 ± 0.07 (%) | [39] |

**Table 2.** *Cont.*

| Sample | Variety/Origin | Treatment | Total Phenols | Total Flavonoids | Total Anthocyanins | Antioxidative Capacity | | References |
|---|---|---|---|---|---|---|---|---|
| "Shir Anjir" | Iran | (13/0) [e] | No tested | No tested | No tested | | No tested | [40] |
| "Shir Anjir" | Iran | (16.5/0) [e] | No tested | No tested | No tested | | No tested | [40] |
| "Shir Anjir" | Iran | (20/0) [e] | No tested | No tested | No tested | | No tested | [40] |
| "Shir Anjir" | Iran | (13/0.35) [e] | No tested | No tested | No tested | | No tested | [40] |
| "Shir Anjir" | Iran | (16.5/0.35) [e] | No tested | No tested | No tested | | No tested | [40] |
| "Shir Anjir" | Iran | (20/0.35) [e] | No tested | No tested | No tested | | No tested | [40] |
| "Shir Anjir" | Iran | (13/0.7) [e] | No tested | No tested | No tested | | No tested | [40] |
| "Shir Anjir" | Iran | (16.5/0.7) [e] | No tested | No tested | No tested | | No tested | [40] |
| "Shir Anjir" | Iran | (20/0.7) [e] | No tested | No tested | No tested | | No tested | [40] |
| Power fig | Iran | FP 707 [f] | No tested | No tested | No tested | | No tested | [41] |
| Power fig | Iran | FP 505 [f] | No tested | No tested | No tested | | No tested | [41] |
| Powerfig | Iran | FP 354 [f] | No tested | No tested | No tested | | No tested | [41] |
| Powder figs | Cuello dama | Peel | $4.78 \pm 0.28$ (mg GAE g$^{-1}$) | $17.61 \pm 0.45$ (mg RE g$^{-1}$) | $6.21 \pm 0.28$ (mg CGE g$^{-1}$) | DPPH | $9.50 \pm 0.11$ (%) [g] | [42] |
| Powder figs | Colar | Peel | $5.76 \pm 0.13$ (mg GAE g$^{-1}$) | $19.12 \pm 0.04$ (mg RE g$^{-1}$) | $16.63 \pm 0.85$ (mg CGE g$^{-1}$) | DPPH | $21.48 \pm 0.53$ (%) [g] | [42] |
| Powder figs | Cuello dama | Pulp | $2.67 \pm 0.01$ (mg GAE g$^{-1}$) | $13.51 \pm 0.14$ (mg RE g$^{-1}$) | nd | DPPH | $8.98 \pm 0.08$ (%) [g] | [42] |
| Powder figs | Colar | Pulp | $1.92 \pm 0.07$ (mg GAE g$^{-1}$) | $9.24 \pm 0.22$ (mg RE g$^{-1}$) | nd | DPPH | $4.12 \pm 0.10$ (%) [g] | [42] |
| Smoothie | Colar | 40% F + 60% Mo [h] | nd | nd | 43.1 (mg/100 g fw) | ABTS | 0.68 (mmol Trolox/100 g fw) | [43–45] |
| Smoothie | Colar | 40% F + 60% W [h] | nd | nd | 75 (mg/100 g fw) | ABTS | 1.01 (mmol Trolox/100 g fw) | [43–45] |
| Smoothie | Colar | 60% F + 40% Mo [h] | nd | nd | 41.9 (mg/100 g fw) | ABTS | 0.72 (mmol Trolox/100 g fw) | [43–45] |
| Smoothie | Colar | 60% F + 40%W [h] | nd | nd | 50.8 (mg/100 g fw) | ABTS | 0.66 (mmol Trolox/100 g fw) | [43–45] |
| Wine | Brown turkey | HT-winess [i] | $651 \pm 12$ (mg L$^{-1}$) | $126 \pm 2$ (mg L$^{-1}$) | $5.9 \pm 0.5$ (mg L$^{-1}$) | DPPH | $22.4 \pm 0.9$ (%) [j] | [46] |
| Wine | Brown turkey | Co-winess [i] | $679 \pm 9$ (mg L$^{-1}$) | $135 \pm 6$ (mg L$^{-1}$) | $6.5 \pm 0.3$ (mg L$^{-1}$) | DPPH | $31.9 \pm 1.3$ (%) [j] | [46] |
| Wine | Brown turkey | HT-winedf [i] | $705 \pm 15$ (mg L$^{-1}$) | $110 \pm 7$ (mg L$^{-1}$) | $3.0 \pm 0.2$ (mg L$^{-1}$) | DPPH | $25.2 \pm 1.2$ (%) [j] | [46] |
| Wine | Brown turkey | Co-winedf [i] | $731 \pm 9$ (mg L$^{-1}$) | $116 \pm 5$ (mg L$^{-1}$) | $3.0 \pm 0.1$ (mg L$^{-1}$) | DPPH | $29.6 \pm 0.8$ (%) [j] | [46] |
| Wine | Hunan, China | WA:PF 1:7 [k] | $725.58 \pm 11.45$ (mg L$^{-1}$) | $124.39 \pm 3.36$ (mg L$^{-1}$) | $148.94 \pm 2.67$ (mg L$^{-1}$) | DPPH | $88.21 \pm 0.23$ (%) | [47] |
| Wine | Hunan, China | PF:HU 3:1 [k] | $682.67 \pm 16.13$ (mg L$^{-1}$) | $180.7 \pm 1.79$ (mg L$^{-1}$) | $115.17 \pm 4.96$ (mg L$^{-1}$) | DPPH | $84.65 \pm 0.54$ (%) | [47] |
| Wine | Hunan, China | WA:HU 7:1 [k] | $744.07 \pm 9.81$ (mg L$^{-1}$) | $143.58 \pm 2.67$ (mg L$^{-1}$) | $116.80 \pm 1.35$ (mg L$^{-1}$) | DPPH | $86.51 \pm 0.42$ (%) | [47] |
| Wine | Hunan, China | WA:PF:HU 3:1:3 [k] | $765.20 \pm 5.51$ (mg L$^{-1}$) | $158.07 \pm 0.71$ (mg L$^{-1}$) | $142.37 \pm 3.72$ (mg L$^{-1}$) | DPPH | $88.65 \pm 0.10$ (%) | [47] |
| Wine | Hunan, China | Saccharomyces 1012 [k] | $735.86 \pm 8.15$ (mg L$^{-1}$) | $115.74 \pm 0.76$ (mg L$^{-1}$) | $90.49 \pm 0.70$ (mg L$^{-1}$) | DPPH | $86.37 \pm 0.34$ (%) | [47] |
| Wine | Hunan, China | uninoculated [k] | $538.35 \pm 26.65$ (mg L$^{-1}$) | $112.72 \pm 1.84$ (mg L$^{-1}$) | $52.50 \pm 2.21$ (mg L$^{-1}$) | DPPH | $6.62 \pm 0.23$ (%) | [47] |

[a] Storage period (months), [b] ground figs, [c] half cut fig, [d] Wheat flour was replaced by fig seed powder at levels of 0%, 5%, 10% and 15%, [e] Dried fig (13%, 16.5% and 20%) and carboxymethylcellulose (CMC) (0%, 0.35% and 0.7%), [f] FP: Fig Powder; 707, 505 and 354: different particle size of fig powder based on micrometer-sized particles, [g] 10 mg/mL of sample, [h] Mo, Mollar de Elche pomegranate juice; W, Wonderful pomegranate juice; F: Fig purée, [i] Fig wine with/without prefermentation heating (HT-wineff/Co-wineff), and dried fig wine with or without prefermentation heating (HT-winedf/Co-winedf), [j] 50 mg/L of gallic acid, [k] Yeast strain/ Inoculation proportion/Abbreviations: HU, *Hanseniaspora uvarum*; PF, *Pichia fermentans*; WA, *Wickeramomyces anomala*.

The bioactive compound content and antioxidant activity strongly depend on the cultivar type in both fresh and processed fruits [48]. Khadhraou et al. [36] studied the main phenolic compounds, as well as the phenolic profiles and antioxidant activity, in nine sun-dried fig cultivars with different skin colors, originating from South-Eastern and Middle Eastern Tunisia [36]. For all evaluated parameters, a considerable variability with high significant differences was observed among the cultivars studied and the principal component analysis showed three groups of cultivars based on their similarity level. Dark cultivars contained the highest levels of flavonoids and phenolics and exhibited a high antioxidant capacity, while light-skinned cultivars contained the lowest levels. A recent study suggest that the preparation of fig jam preserves some bioactive compounds, especially carotenoids and phenolic compounds during storage [49]. On the other hand, Rababah et al. [33] studied the total phenolics and anthocyanins of fig jam after five months of storage and concluded that jam processing decreased the total phenolics (by 68.6%) and anthocyanins (by 60.2%). The minimum value to total phenolics and anthocyanins was 130.97 mg GAE kg$^{-1}$ and 11.20 mg kg$^{-1}$ of cyanidin-3-glucoside, respectively (Table 2).

As for dried figs, Slatnar et al. [34] showed results of total phenolics after a drying treatment. The drying process affected the degradation of phenolic compounds, the content of phenolic compounds being higher in fresh figs, followed by oven-dried figs and sun-dried figs. For example, Vallejo et al. [5] showed that around 15% of the total phenolics were lost in the drying processes in figs "Cuello Dama". Not only is the quality important, but safety is essential to be maintained. Mycotoxins have been found in quantities above the recommended limit in commercial samples of dry figs [50]. Therefore, a controlled drying process helps to reach a safety level. Alternatives to traditional sun drying is necessary for improving the protection of public health [51].

Nowadays, there is an increase on developing nutrient-rich value-added products by partially replacing its ingredients by others, such as underutilized fruits and added value by-products (pectins, colorants, emulsifier and antioxidants) from leaves and peels. As for fig by-products, Table 2 shows the reported inclusion alternatives; for example, fig powder as a colorant in the production of buns and muffins [52]. The authors also reported how the addition of fig seed powder to the formulation of a cookie improved its fiber content and also increased the total phenolic content and antioxidant activity [39]. Additionally, fig by-products' sweet extracts have been used for making traditional desserts without adding sugar, for example "Shir Anjir", an Iranian dessert [40]. Minimally processed fruits, such as smoothies, retain a large number of phytochemicals and they could in fact be considered a valid alternative to eating fresh fruits. Moreover, De Pilli et al. [53] found that the polyphenol content and antioxidant activity are strongly correlated in both fresh fruits and smoothies. In the same way, fig and pomegranate smoothies also showed a correlation between anthocyanins content and antioxidant capacity; smoothies with 60% of wonderful pomegranate juice showed a higher anthocyanin and antioxidant capacity (Table 2) [43–45]. Other studies have also suggested that fig leaf extracts presented a potential use as a source of natural antioxidant compounds [54]. Lu et al. [3] has noted that dried fig wines had lower contents of anthocyanins than fresh fig wines, which could be because of the thermal degradation of anthocyanins during the fig drying process. These wines also showed a lower antioxidant capacity.

Recent studies suggested that by-products/co-products obtained from peel and fig pulp showed potential properties to be used as ingredient in food products/additives (Tables 1–3). Table 3 shows the reported research about different raw fig by-product materials (different plant parts, peel, leaves and whole figs) and the extraction method used to obtain the desired ingredients/additives and their uses. For instance, peel extract could be used as a colorant due to its potential source of anthocyanin. Consequently, fig peel extract has great potential to be used as a natural food dye, where in addition to its ability to add natural purple colors, it also presents interesting antioxidant and antimicrobial activities. Table 3 also shows the extraction and uses of pectin from fig peel and pulp [55].

**Table 3.** Different potential uses of underutilized fruits and extracts of fig by-products.

| Plant Part | Extract | Method | Uses | References |
|---|---|---|---|---|
| **Peel** | Lyophilized powdered | Extracted with 100 mL of acidified solvent 100% etanol | Natural purple colorants | [56] |
| | Lyophilized powdered | Heat-assisted extraction Microwave-assisted extraction Ultrasound-assisted extraction | Bioactive anthocyanin pigments | [57] |
| | Pectin | Hot-water extraction Ultrasound-assisted extraction Microwave-assisted extraction | The strong antioxidant and emulsification capacities | [58] |
| **Leaves** | Aqueous extract | Finely ground leaf powder suspended in 96 mL deionized water filtered by sterilized membrane filter, concentrated by using a rotary evaporator at 50 °C and followed by drying in an oven at 50 °C | Prolong the shelf life of pasteurized milk | [31] |
| | Powdered | Ethanol and chloroform were used as extracting solvents | Milk-clotting activity, which is most likely due to an enzyme component | [59] |
| | Fresh Leaf | Fig leaf extract, 96% ethanol. Using the maceration method | Antibacterial activity of fig leaf extracts | [60] |
| | Powdered | 10 g of the finely divided leaf particles was dissolved in 200 mL of deionized water in a 500 mL flat bottom flask | Synthesis of eco-friendly and sustainable nanoparticles | [61] |
| | Fresh leaves and stems of the wild fig | Simple and chemical-free method (crushed and centrifuged). | Clotting ability in goat's fresh cheese production | [62] |
| | Powdered | Surfactant (PEG8000)-based microwave-assisted extraction method | Source of bioactive compounds | [63] |
| | Powdered | 0.1 g of sample and 10 mL aqueous 50% acetone, centrifuged using Eppendorf centrifuge and filtered with a 0.22 µm PTFE syringe filter. | Source of bioactive compounds | [54] |
| **Whole figs** | Syrup | 100 g of low-quality dried fig fruits were soaked in 500 mL distilled water, mixed and then centrifuged to remove solids. | Pullulan gum production from low-quality fig syrup using Aureobasidium pullulans | [64] |
| | Powdered | Samples (1 g) were mixed with ethanol (50 mL) and left macerating for 24 h; then, solutions were centrifuged ($6800\times g$/20 min) and extraction was repeated three times. | Source of bioactive compounds | [65] |
| | Dry fig and stevia extract | Microwave-assisted extraction of stevia | Sugar replacement in ice cream | [66] |

Regarding leaf extracts, El Dessouky Abdel-Aziz et al. [31] suggested that they can be used to extend the shelf life of pasteurized milk from 5 to 16 days without altering organoleptic properties. Moreover, other authors have reported that leaf extracts or fig powder can be a potential product for manufacturing functional foods [46] (Table 3). Fermentation is also known to promote the concentration of bioactive compounds of fruits and vegetables [55].

## 5. Conclusions

Although there has been an increase in research focused on the bioactive compounds of fig fruits and their by-products, more scientific evidence (combined with a unified way

of publishing data on bioactive compound content) is needed to establish the potential health properties. Future investigations should be focused on in vitro and in vivo studies to reveal their beneficial properties. There is scientific research about the potential use of underutilized fig fruit and figs by-products and its bioactive compounds as nutritional, functional and techno-functional properties. The use and valorization of the waste material (leaves, peel and pulp) produced during fig processing should be further investigated, since this could offer financial benefits to farmers and solve environmental issues by ensuring the sustainable management of these materials and, furthermore, bringing benefits to consumers' health and well-being. In addition, an economic estimation of the bioactive compounds of fig by-products could be essential to gain more knowledge and obtain added value. Although fig-based products and their uses were reported, such as smoothies, fig powders, colorants, fermented drinks and biscuits, among others, in the future, other products should be researched, for instance: fig coffee, dried figs using novel technologies and fermented milks based on fig by-products.

**Author Contributions:** Conceptualization, M.C.-L. and F.H.; Methodology, M.C.-L., E.S. and F.H.; Investigation, C.T.-A.; Writing—Original Draft Preparation, C.T.-A. and L.A.-C.; Writing—Review and Editing, D.L.-L. and E.S.; Visualization, D.L.-L.; Supervision, M.C.-L., E.S., D.L.-L. and F.H. All authors have read and agreed to the published version of the manuscript.

**Funding:** This research received no external funding.

**Institutional Review Board Statement:** Not applicable.

**Informed Consent Statement:** Not applicable.

**Data Availability Statement:** No applicable.

**Conflicts of Interest:** The authors declare no conflict of interest.

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
