# Peer review of "Ficus carica Fruits, By-Products and Based Products as Potential Sources of Bioactive Compounds: A Review"

_agronomy, doi:10.3390/agronomy11091834_

Round 1
Reviewer 1 Report
The authors have revised the requested items. As recommended they have added more related references and previously published reports. The present form might be interesting for scholar readers, so it can be considered for further publication procedure.
Reviewer 2 Report
I am of opinion that the manuscript has been significantly improved and is now ready for the publication in Agronomy.
This manuscript is a resubmission of an earlier submission. The following is a list of the peer review reports and author responses from that submission.
Round 1
Reviewer 1 Report
Dear authors,
Although, the manuscript is well-written, the novelty is a crucial concern. By a simple search, several review papers can be found focusing on various Ficus carica parts: https://doi.org/10.3109/13880209.2014.892515, https://doi.org/10.1155/2013/974256; And a recently published https://doi.org/10.1016/j.biopha.2021.111393 which reviews "phytochemistry, bioactivities, toxicity studies, and clinical studies on Ficus carica Linn. leaves", whereas they are not mentioned in the manuscript. Although, it is the first report on the fig's food products, in my opinion there is no sufficient data to review.
Moreover, the study does not comprehensively overview the experiments performed on the plant parts, 55 references for this broad topic is not enough; besides the chemical structures of the identified components are not drawn.
Regarding the abovementioned reasons, in my opinion the manuscript has major drawbacks.
Author Response
Reviewer 1
- Although, the manuscript is well-written, the novelty is a crucial concern. By a simple search, several review papers can be found focusing on various Ficus carica parts: https://doi.org/10.3109/13880209.2014.892515 https://doi.org/10.1155/2013/974256 And a recently published https://doi.org/10.1016/j.biopha.2021.111393 which reviews "phytochemistry, bioactivities, toxicity studies, and clinical studies on Ficus carica Linn. leaves", whereas they are not mentioned in the manuscript. Although, it is the first report on the fig's food products, in my opinion there is no sufficient data to review. Moreover, the study does not comprehensively overview the experiments performed on the plant parts, 55 references for this broad topic is not enough; besides the chemical structures of the identified components are not drawn. Regarding the above mentioned reasons, in my opinion the manuscript has major drawbacks.
Thanks for the kind comment that has been considered by the authors to improve the manuscript. Authors checked the recommended literature, and the information was added in the last version of the manuscript. Authors agree with the editor and the reviewers that some important manuscripts were missed, then authors made a larger review, incorporating the last articles related to this review (n = 12). In the current version, a total of 67 manuscripts were added in both the main text (example: introduction and discussion section: 124-134) and the tables (Table 1, 2 and 3), changes were added in blue color. The main revision, articles which were included in tables and were discussed were 41 as updated methodology section shows. Some changes in the text were done for updating the information (Line 111). As to chemical structures, authors agree with the reviewer that it can be interesting. Authors incorporated a sentence in lines 114-116 indicating that the main chemical structure of fig and derivatives are shown in two recent manuscripts: “The chemical structure of the main compounds found in fig fruits and derivatives are shown in recent studies related to chemical composition [14, 16]”.

Reviewer 2 Report
Dear authors, the paper it's very interesting. However, I would like to give some suggestions to improve your manuscript.
1) Page 7 line 137. It is advisable to add some information and discussion about the cultivar effect on the content of polyphenols compounds and antioxidant activity. I will suggest inserting the following sentences.
Actually, the bioactive compound content and antioxidant activity strongly depend on the cultivar type in both fresh or processed fruits (De Pilli et al., 2019). Khadhraou et al. (2019) studied the main phenolic compounds, as well as phenolic profiles and antioxidant activity in nine sun-dried fig cultivars with different skin colours, originating from South-Eastern and MiddleEastern Tunisia. For all evaluated parameters, a considerable variability with high significant differences was observed among the cultivars studied and the principal component analysis showed three groups of cultivars regarding their similarity level. Dark cultivars contained the
highest levels of flavonoids and phenolics and exhibited high antioxidant capacity, while light-skinned cultivars contained the lowest levels.
2) Page 8 line 165. I suggest enriching this information and to change this sentence in the following way.
Minimally processed fruits, like smoothies, retain a large number of phytochemicals and they could actually be considered a valid alternative to fresh eaten fruits. Moreover, De Pilli and Lopriore (2018) found that the polyphenol content and antioxidant activity are strongly correlated in both fresh fruits and smoothies. In the same way, fig and pomegranate smoothies also showed a correlation between anthocyanins content and antioxidant capacity, smoothies with 60% of wonderful pomegranate juice were the ones that showed higher anthocyanin and antioxidant capacity (Table 2) [39, 40, 41].
References
1) De Pilli, T., Lopriore, G., Montemitro M., Alessandrino O. (2019). Effects of two sweet cherry cultivars (Prunus avium L., cvv. ‘Ferrovia’ and ‘Lapins’) on the shelf life of an innovative bakery product. J Food Sci Technol 56(1):310–320. https://doi.org/10.1007/s13197-018-3491-5
2) Khadhraoui, M., Bagues, M., Artés, F., Ferchichi, A. (2019). Phytochemical content, antioxidant potential, and fatty acid composition of dried Tunisian fig (Ficus carica L.) cultivars. Journal of Applied Botany and Food Quality 92, 143 - 150. DOI:10.5073/JABFQ.2019.092.020
3) De Pilli, T. and Lopriore G. (2018). Ripeness stage effects on quality characteristics of smoothies made up of sweet cherries (P. AviumL., cv. ‘Lapins’). Emirates Journal of Food and Agriculture. 30(11): 959-967
doi: 10.9755/ejfa.2018.v30.i11.1861
Author Response
Reviewer 2
- Dear authors, the paper it's very interesting. However, I would like to give some suggestions to improve your manuscript.
Authors thank the reviewer for this comment. All suggestions have been taken into consideration to improve the manuscript, they are in blue color.
- Page 7 line 137. It is advisable to add some information and discussion about the cultivar effect on the content of polyphenols compounds and antioxidant activity. I will suggest inserting the following sentences.
“Actually, the bioactive compound content and antioxidant activity strongly depend on the cultivar type in both fresh or processed fruits (De Pilli et al., 2019). Khadhraou et al. (2019) studied the main phenolic compounds, as well as phenolic profiles and antioxidant activity in nine sun-dried fig cultivars with different skin colours, originating from South-Eastern and Middle-Eastern Tunisia. For all evaluated parameters, a considerable variability with high significant differences was observed among the cultivars studied and the principal component analysis showed three groups of cultivars regarding their similarity level. Dark cultivars contained the highest levels of flavonoids and phenolics and exhibited high antioxidant capacity, while light-skinned cultivars contained the lowest levels”
Authors agree with this comment. The suggested sentences were added. Please, see lines 145-155.
- Page 8 line 165. I suggest enriching this information and to change this sentence in the following way.
“Minimally processed fruits, like smoothies, retain a large number of phytochemicals and they could actually be considered a valid alternative to fresh eaten fruits. Moreover, De Pilli and Lopriore (2018) found that the polyphenol content and antioxidant activity are strongly correlated in both fresh fruits and smoothies. In the same way, fig and pomegranate smoothies also showed a correlation between anthocyanins content and antioxidant capacity, smoothies with 60% of wonderful pomegranate juice were the ones that showed higher anthocyanin and antioxidant capacity (Table 2) [39, 40, 41]”.
Authors agree with this comment. The suggested sentences were added. Please, see lines 177-180. Also the suggested references were added in the text and/or in Tables:
- De Pilli, T., Lopriore, G., Montemitro M., Alessandrino O. (2019). Effects of two sweet cherry cultivars (Prunus avium L., cvv. ‘Ferrovia’ and ‘Lapins’) on the shelf life of an innovative bakery product. J Food Sci Technol 56(1):310–320. https://doi.org/10.1007/s13197-018-3491-5
- Khadhraoui, M., Bagues, M., Artés, F., Ferchichi, A. (2019). Phytochemical content, antioxidant potential, and fatty acid composition of dried Tunisian fig (Ficus carica L.) cultivars. Journal of Applied Botany and Food Quality 92, 143 - 150. DOI:10.5073/JABFQ.2019.092.020
- De Pilli, T. and Lopriore G. (2018). Ripeness stage effects on quality characteristics of smoothies made up of sweet cherries (P. AviumL., cv. ‘Lapins’). Emirates Journal of Food and Agriculture. 30(11): 959-967 doi: 10.9755/ejfa.2018.v30.i11.1861
Also, authors agree with the editor and the reviewers that some important manuscripts were missed, then authors made a larger review, incorporating the last ones related to this review (n = 12). In the current version, a total of 67 manuscripts were added.

Reviewer 3 Report
The manuscript is a short review on the phenolic composition of different parts of figs, including the by-products and some fig based products. I would like to complement the Authors for the work.
The article is well written and interesting. However, there are several aspects to consider:
Abstract
- Lines 11-12: I would suggest changing a bit the 1st sentence to be as following: “In this review, studies (n=32) were searched on which compounds and their content have been found in whole fig fruit, peel, leaves and pulp, the types of fig based products and ON their total phenols and antioxidant capacity AS WELL AS on the potential uses of different extracts of fig parts.”
- Line 13: Did you mean “zero waste” instead of “waste cero”? It is mentioned also in the text (line 83).
- Lines 18-21: The values of the individual phenolic compounds mentioned are quite detailed. Are they just from 1 study? As this is a review and not research article reporting results, I would suggest removing these values from the abstract and leaving just the mentioned compounds.
- Lines 21-24: Please, correct these 2 sentence to be as following: “A potential strategy could be the development OF novel additives and/or ingredients for food industry from fig BY-PRODUCTS. Therefore, the use and valorization OF waste material produced during figs processing should be further investigated.”
Keywords: You focused on fig by-products, so I would suggest removing the last word (co-product)
- Introduction
- Line 28: Please, correct the word “specie” to “species”
- In the whole text you are using both Ficus carica L. and fig. I would suggest mentioning the Latin name at the beginning of the article and then continuing with “fig”.
- Lines 39-40: Please, add a reference for this sentence.
- Line 44: Check and correct this sentence.
- Line 46-47: I would suggest adding the reference for the sentence about the fig production in Spain.
- Line 52: “…for his higher content of sugars.” Did you mean: “…due to their higher content of sugars”?
- Lines 56-58: Respectively to what? Also, please, if you mention “several authors”, then put all the references and not just 1 (the same also later in the text, lines 141-142).
- Lines 61-62: I would suggest changing the sentence to something like: “There is just one authorized claim (for polyphenols): hydroxytirosol and derivatives in olive oil.”
- Lines 80-81: Please, correct this sentence to be as following: “Similar results were also reported in a previous study in WHICH fig leaf extracts WERE USED TO help combat eating and lifestyle disorders.”
- Lines 89-90: I am of opinion that the objective of this review is not as much the old, present and future uses of products but more how the processing (drying, preparation of jams) and storage have affected the phenolic composition of fig products.
- Regarding the subtitles, I would suggest changing them a bit (be careful to number them correctly (you have put number 3 to both of the subtitles (line 112 and line 133)).
Maybe to put something like: “3. Bioactive compounds in different fig parts” and for the next one “4. Bioactive content of fig based products and their antioxidant activity”
or
“3. Phenolic composition of different fig parts” and “4. Phenolic content and antioxidant activity of fig based products”
- Lines 113-114: Please, correct the following: “…found in different FIG partS: whole fruit (n=19)….”
- In the text of the Chapter 3 you mention just the highest values reported and presented in the table 1. I am of opinion that you could or remove the values from the text (as you have them in the table) or to put the whole range of the reported contents of individual compounds (as the values are not from the same study but from different ones, so not completely comparable, as you state in the text).
- Lines 134-135: Please, rewrite the sentence to be a bit clearer to understand.
- Lines 136-138: I would suggest changing a bit the sentence: “Table 2 shows important information (type, cultivar, treatment) and bioactive compounds (total phenols, total flavonoids, total anthocyanins) as well as antioxidant capacity of products based on fig fruits and by-products.”
- Please, in the whole manuscript write the units in the same way (according to the Guide for Authors of the Journal): line 146, Table 2
- Line 154: “…in commercial samples OF dry figs.”
- Line 157: Who is working on it? The beginning of the sentence is not clear.
- Line 163: Which fig by-product?
- Line 167: Wonderful
- Line 169: “…presented a potential USE as a source of natural antioxidant compounds.”
- Line 170: “Lu et al. (3) HAS noted that dried fig wines were lower IN content OF anthocyanins than fresh fig wines….”
Table 2: Please correct the word “origen” to “origin”
Table 2: Write all the units in the same way
Table 2: Did you mean “powder fig” for the study of Yeganehzad et al., 2020 (46)? For the same study, what was measured there? Were the total phenols, total flavonoids and total anthocyanins tested but then not detected?
- Line 182: “…(different PLANT PARTS, peel, leaves and whole figs)…”
- Line 188: “…extracts can be used TO extend the shelf life….”
- Line 190: “Moreover, OTHER authors have reported…”
Table 3: I would suggest correcting the title to: “Different potential uses of underutilized fruits and extracts of fig by-products”
Table 3: “Plant part” instead of “Part plant”
Table 3: Once again, did you mean “powdered” instead “powered”?
- Conclusions
- Line 198: “….publishing data ON bioactive compounds content) are needed to ESTABLISH the potential…”
- Line 206: “In addition, an economic ESTIMATION of the bioactive compounds of fig by-products could be essential to gain more knowledge and obtain added value. Although fig based products and THEIR uses were reported (such as smoothies, fig powders, colorants, fermented drinks, biscuits, among others), in the future, other products should be researched…..”
Author Response
Reviewer 3
- The manuscript is a short review on the phenolic composition of different parts of figs, including the by-products and some fig based products. I would like to complement the Authors for the work. The article is well written and interesting. However, there are several aspects to consider
Authors thank for the comments, they will improve the version of the manuscript.
Abstract
- Lines 11-12: I would suggest changing a bit the 1stsentence to be as following: “In this review, studies (n=32) were searched on which compounds and their content have been found in whole fig fruit, peel, leaves and pulp, the types of fig based products and ON their total phenols and antioxidant capacity AS WELL AS on the potential uses of different extracts of fig parts.”
Suggestion was added (Line 12).
- Line 13: Did you mean “zero waste” instead of “waste cero”? It is mentioned also in the text (line 83).
Zero waste replaced waste cero in Abstract and in the text. Thank you for this comment.
- Lines 18-21: The values of the individual phenolic compounds mentioned are quite detailed. Are they just from 1 study? As this is a review and not research article reporting results, I would suggest removing these values from the abstract and leaving just the mentioned compounds.
The values were from the study in which the content was the highest one. Authors removed the values as reviewer suggested.
- Lines 21-24: Please, correct these 2 sentence to be as following: “A potential strategy could be the development OF novel additives and/or ingredients for food industry from fig BY-PRODUCTS. Therefore, the use and valorization OF waste material produced during figs processing should be further investigated.”
Changes were updated.
Keywords:
- You focused on fig by-products, so I would suggest removing the last word (co-product)
Authors remove this term and added-value was incorporated to keywords.
Introduction
- Line 28: Please, correct the word “specie” to “species”
Done as suggested.
- In the whole text you are using both Ficus carica and fig. I would suggest mentioning the Latin name at the beginning of the article and then continuing with “fig”.
Authors agree with the suggestion. Authors reviewed the text and Latin name was replaced to fig.
- Lines 39-40: Please, add a reference for this sentence.
Sorry for missing this essential reference. Authors incorporated it in the current version of the review.
- Line 44: Check and correct this sentence. In Spain, the main producer is Extremadura (28,749 t), followed by Cataluña (5,978 t) and the Comunidad Valenciana.
Sorry for this mistake. Production of Comunidad Valenciana was added and the sentence was updated. Updated statisticals from Spanish Goverment (2020) was added and the reference was added. Line 43-44.
- Line 46-47: I would suggest adding the reference for the sentence about the fig production in Spain.
Sorry for missing this essential reference. Authors incorporated it in the current version of the review.
- Line 52: “…for his higher content of sugars.” Did you mean: “…due to their higher content of sugars”?
Sorry for this mistake. Line 50, mistake was solved.
- Lines 56-58: Respectively to what? Also, please, if you mention “several authors”, then put all the references and not just 1 (the same also later in the text, lines 141-142).
Firsly, recent and suggested refereces were added (Line 53-55), while in the second suggestion authors realice that the comment was only related to one study (Lines 153-155).
- Lines 61-62: I would suggest changing the sentence to something like: “There is just one authorized claim (for polyphenols): hydroxytirosol and derivatives in olive oil.”
Suggestion was added. Lines 58-59.
- Lines 80-81: Please, correct this sentence to be as following: “Similar results were also reported in a previous study in WHICH fig leaf extracts WERE USED TO help combat eating and lifestyle disorders.”
Mistakes were solved. Thanks.
- Lines 89-90: I am of opinion that the objective of this review is not as much the old, present and future uses of products but more how the processing (drying, preparation of jams) and storage have affected the phenolic composition of fig products.
Authors agree with this comment. Check lines 86-87
- Regarding the subtitles, I would suggest changing them a bit (be careful to number them correctly (you have put number 3 to both of the subtitles (line 112 and line 133). Maybe to put something like: “3. Bioactive compounds in different fig parts” and for the next one “4. Bioactive content of fig based products and their antioxidant activity” or “3. Phenolic composition of different fig parts” and “4. Phenolic content and antioxidant activity of fig based products”.
Authors changed the subtitles and changed the numbers as follows: “3. Bioactive compounds in different fig parts” and for the next one “4. Bioactive content of fig based products and their antioxidant activity”.
- Lines 113-114: Please, correct the following: “…found in different FIG partS: whole fruit (n=19).”
Done as suggested (Line 111).
- In the text of the Chapter 3 you mention just the highest values reported and presented in the table 1. I am of opinion that you could or remove the values from the text (as you have them in the table) or to put the whole range of the reported contents of individual compounds (as the values are not from the same study but from different ones, so not completely comparable, as you state in the text).
Authors removed the values as reviewer suggested. Line 117-123.
- Lines 134-135: Please, rewrite the sentence to be a bit clearer to understand.
Authors rewrote the beggining of the paragraph.
- Lines 136-138: I would suggest changing a bit the sentence: “Table 2 shows important information (type, cultivar, treatment) and bioactive compounds (total phenols, total flavonoids, total anthocyanins) as well as antioxidant capacity of products based on fig fruits and by-products.”
Done as suggested. Lines 140-142.
- Please, in the whole manuscript write the units in the same way (according to the Guide for Authors of the Journal): line 146, Table 2
Units were updated in text (Line 158-159) and Table 2: /g, /kg and /L was changed to g-1 Kg-1 and L-1.
- Line 154: “…in commercial samples OF dry figs.”
Done as suggested.
- Line 157: Who is working on it? The beginning of the sentence is not clear.
The begginis of the sentence was re-written. Line 169.
- Line 163: Which fig by-product?
Authors clarified in the original versión in the sentence before that ig by-products means “pectins, colorants, emulsifier, antioxidants”, and authors authors in the currrent version that this compounds are from leaves, under-utilized fruits, and peels. Line 171.
- Line 167: Wonderful
Done as suggested. Line 183.
- Line 169: “…presented a potential USE as a source of natural antioxidant compounds.”
Done as suggested.
- Line 170: “Lu et al. (3) HAS noted that dried fig wines were lower IN content OF anthocyanins than fresh fig wines”
Sorry for this English mistakes. They were solved.
- Line 182: “…(different PLANT PARTS, peel, leaves and whole figs)…”
Done as suggested.
- Line 188: “…extracts can be used TO extend the shelf life….”
Done as suggested.
- Line 190: “Moreover, OTHER authors have reported…”
Done as suggested.
- Did you mean “powder fig” for the study of Yeganehzad et al., 2020 (46)? For the same study, what was measured there? Were the total phenols, total flavonoids and total anthocyanins tested but then not detected?
Authors agree with the reviewer, this information was not completed. The current version indicates that the total phenols, flavonoids and anthocyanins were no quantitatively tested (reference 43 and 53 in Table 2) but authors decided to include these references due to the topic of the manuscripts and the Table. To avoid misunderstanding, authors changed nd (no detected) to no tested.
Table 2
- Please correct the word “origen” to “origin”
Done as suggested.
- Write all the units in the same way
Units were updated. /g and /L was changed to g-1 and L-1 as Table 1 shows.
Table 3
- I would suggest correcting the title to: “Different potential uses of underutilized fruits and extracts of fig by-products”
Done as suggested.
- “Plant part” instead of “Part plant”
Done as suggested.
- Once again, did you mean “powdered” instead “powered”?
Done as suggested.
Conclusions
- Line 198: “publishing data ON bioactive compounds content) are needed to ESTABLISH the potential…”
Done as suggested.
- Line 206: “In addition, an economic ESTIMATION of the bioactive compounds of fig by-products could be essential to gain more knowledge and obtain added value. Although fig based products and THEIR uses were reported (such as smoothies, fig powders, colorants, fermented drinks, biscuits, among others), in the future, other products should be researched…..”
Done as suggested.
